# LASER: Attention with Exponential Transformation

**Sai Surya Duvvuri** [1]   **Inderjit S. Dhillon** [2]

## Abstract

Transformers have had tremendous impact for several sequence related tasks, largely due to their ability to retrieve from any part of the sequence via softmax based dot-product attention. This mechanism plays a crucial role in Transformer's performance. We analyze the gradients backpropagated through the softmax operation in the attention mechanism and observe that these gradients can often be small. This poor gradient signal backpropagation can lead to inefficient learning of parameters preceeding the attention operations. To this end, we introduce a new attention mechanism called LASER, which we analytically show to admit a larger gradient signal. We show that LASER attention can be implemented by making small modifications to existing attention implementations. We conduct experiments on autoregressive large language models (LLMs) with upto 7.7 billion parameters with an average improvement of upto 1.44% over standard attention on downstream evaluations and 1.65% finetuning improvements. Additionally, LASER demonstrates generalization performance improvement across a variety of tasks (vision, text and speech):Vision Transformer (ViT) on Imagenet, Conformer on the Librispeech speech-to-text and BERT with 2.2 billion parameters.

## 1. Introduction

Transformer architectures (Vaswani et al., 2017) have gained prominence over traditional models like LSTMs (Long-Short-Term-Memory) (Hochreiter & Schmidhuber, 1997) for various sequence-based tasks due to their ability to better capture long-range dependencies (Gemini, 2024; Meta-AI, 2024) without suffering from the vanishing gradient

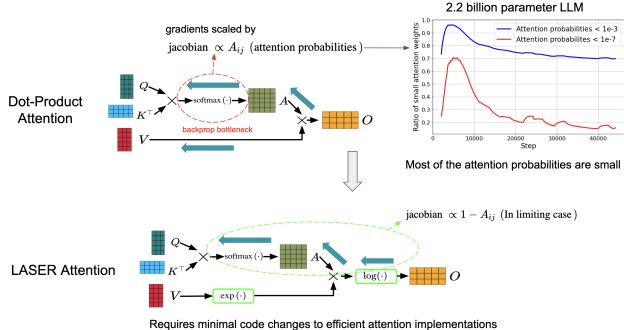

*Figure 1.* Backpropagating gradients through the softmax operation in the attention mechanism requires scaling with the Jacobian of softmax. We show that this Jacobian scales with attention probabilities/weights, which are typically small in large language models (LLMs) with about 80% of the probabilties less than $10^{-3}$ and about 20% less than $10^{-7}$. We propose LASER attention that involves conducting dot-product attention with an $\exp(\cdot)$-transformed value matrix $V$, i.e., conducting attention on $\exp(V)$. We show that LASER admits a larger Jacobian, is easier to implement and does not require any change to the underlying attention function, which may have a more nuanced implementation (e.g., FlashAttention (Dao et al., 2022)). In the image, $\exp(.)$ and $\log(.)$ are element-wise operations.

problem (Glorot & Bengio, 2010; Bengio et al., 1994). The attention mechanism plays a key role in Transformers, where different weights or probabilities are assigned to token representations in a sequence, indicating their relative importance, and these weights are computed via a softmax function (Vaswani et al., 2017). The Transformer architecture consists of multiple stacked layers comprising attention mechanism, where each layer operates on the output of the previous one, forming the Transformer encoder or decoder. Learning within a neural network is performed via gradient backpropagation, wherein gradients propagate backward through the network layer by layer using the chain rule (LeCun et al., 2002). During backpropagation, gradient magnitudes tend to diminish, resulting in a weaker gradient signal reaching the bottom layers and inefficient learning, which is called as vanishing gradient problem (Glorot & Bengio, 2010; Bengio et al., 1994). Residual connections (He et al., 2016) are used in Transformers so that gradients can bypass the layers via skip connections during backpropagation to improve the gradient magnitude for bottom

---

[1]Department of Computer Science, University of Texas at Austin [2]Google. Correspondence to: Sai Surya Duvvuri <saisurya@cs.utexas.edu>.

*Proceedings of the 42$^{nd}$ International Conference on Machine Learning*, Vancouver, Canada. PMLR 267, 2025.

layers, reinforcing the idea that architectures capable of efficient gradient backpropagation tend to offer better training performance.

In this paper, we theoretically analyze the gradient backpropagation in the attention mechanism of a Transformer and identify a vanishing gradient issue. During backpropagation, the gradient can be scaled by a very small value due to the softmax operation in the attention mechanism. Based on this observation, we propose a modification to the attention mechanism - LASER - LogArithm of Summed Exponentials of Representations. LASER is equivalent to conducting attention on exponentially transformed inputs and admits a log-sum-exp structure. We analytically show that gradients propagated via LASER attention are typically large. Since $\exp(\cdot)$ transformation in LASER can lead to numerical overflow, we develop a novel implementation - Log-Weighted-Sum-Exp trick, inspired from the Log-Sum-Exp trick (Blanchard et al., 2019). This technique allows LASER to scale to large models with upto 2.2 billion parameter models. We show that our implementation requires small modifications, and doesn't need any changes to the underlying attention mechanism which might admit a more nuanced implementation, for e.g., FlashAttention (Dao et al., 2022).

We conduct thorough empirical verification across a variety of Transformer models: Conformer (Gulati et al., 2020) for Librispeech speech-to-text (Panayotov et al., 2015), Vision Transformer(Dosovitskiy et al., 2021) for ImageNet classification (Deng et al., 2009), decoder-only text Transformer (Brown et al., 2020) on C4 dataset (Raffel et al., 2020) and BERT ((Devlin et al., 2018)). We conduct experiments on decoder-only autoregressive language models from 234 million parameters to 7.7 billion parameter models, where we demonstate improvements of up to 1.7% relative improvement in test loss over standard attention. We conduct one-shot evaluation on several downstream tasks and show that LASER outperforms standard attention with upto 1.44% accuracy improvement on average and 1.65% improvement on average upon finetuning. On a 2.2 billion parameter BERT (Devlin et al., 2018), LASER gives a relative improvement of 0.93% on masked language modeling prediction error rate. LASER also demonstrates 1.2% improvement in accuracy, and 0.2% improvement in validation word error rate in the Conformer benchmark.

## 2. Related Work

The attention mechanism was used in Bahdanau et al. (2015) to drastically improve machine translation performance compared to encoder-decoder recurrent neural networks (RNNs) (Cho, 2014). This was later adopted in Transformers (Vaswani et al., 2017), which introduced self-attention to improve the performance in machine translation even

further. Efficient attention mechanisms have been an active area of research due to the quadratic computational complexity in sequence length of Attention, which prevents long-context language modeling. One notable contribution is Linear Attention (Katharopoulos et al., 2020), which reduces the quadratic complexity of self-attention to linear in sequence length by using kernel approximation of the softmax operation. Similarly, the Performer (Choromanski et al., 2021) develops an alternative kernel approximation using random feature maps to achieve linear complexity.

A relevant work (Veličković et al., 2024) studies the importance of expressing sharp distributions in attention mechanisms of transformer-based large language models. This helps with downstream tasks such as text retrieval. In contrast, our paper finds that sharp distributions in softmax lead to gradient bottleneck during backpropagation and proposes LASER to fix this issue. In large vision transformer (Dosovitskiy et al., 2021) pretraining, sharp softmax attention distributions led to training instabilities in (Dehghani et al., 2023), the authors propose QK-normalization, which applies layer normalization to queries and keys before computing attention weights. This equilibrates the norms of all the queries and keys, leading to uniform distributions. In (Kim et al., 2021), the authors identified that standard attention mechanism is not $\ell_2$-Lipschitz and develop $\ell_2$ multi-head attention, which uses $\ell_2$-divergence instead of dot-product between key and query vectors. This formulation admits a Lipschitz constant.

## 3. LASER Attention: LogArithm of Summed Exponentials of Representations

We first formally introduce Transformers (Vaswani et al., 2017) and the underlying softmax dot-product attention in Section 3.1. In Section 3.2, we introduce LASER Attention by first deriving the gradients of standard attention by making observations on a simple case of sequence length 2, and then generalizing to larger sequence lengths.

### 3.1. Transformers and Softmax Dot-Product Attention

Let $X \in \mathbb{R}^{N \times d}$ represent the input sequence with $N$ tokens, where the $i$-th row is a $d$-dimensional representation of the $i$-th token. We describe the Transformer layer $T : \mathbb{R}^{N \times d} \to \mathbb{R}^{N \times d}$ similar to (Katharopoulos et al., 2020) as follows:

$$T(X) = f(X + \mathrm{attn}(X)W_O), \tag{1}$$

where $f : \mathbb{R}^{N \times d} \to \mathbb{R}^{N \times d}$ is usually implemented using a 2-layer feed-forward neural network which acts on each token representation independently and $W_O \in \mathbb{R}^{d \times d}$ is a tunable parameter matrix. The attention function $\mathrm{attn}(.)$ is the only operation in the Transformer which is applied across the sequence axis. A single headed attention mecha-

nism (Vaswani et al., 2017) can be described as follows:

$$Q = XW_Q, \quad K = XW_K, \quad V = XW_V,$$
$$\tilde{A} = QK^\top$$
$$\text{attn}(X) = \text{softmax}(\tilde{A})V, \quad (2)$$

where $Q, K, V, \text{attn}(X) \in \mathbb{R}^{N \times d}$. The softmax (Bridle, 1990) operation is applied for each row $\tilde{a}$ of attention logits $\tilde{A} = QK^\top$ separately:

$$(\text{softmax}(\tilde{a}))_i = \exp(\tilde{a}_i) / \left( \sum_{j=1}^{N} \exp(\tilde{a}_j) \right).$$

Layer normalizations (Ba et al., 2016) are usually applied before or after $f(.)$ and $\text{attn}(.)$ (Xiong et al., 2020), but we omit this for brevity. A Transformer is a composition of multiple layers (1) — $T_l(X)$, $l \in \{1, \ldots, L\}$ sandwiched by the input embedding layer $E : \mathbb{R}^{N \times V} \to \mathbb{R}^{N \times d}$ and output softmax layer $S : \mathbb{R}^{N \times d} \to \mathbb{R}^{N \times V}$ as follows:

$$\text{Transformer}(Z) = S \circ T_L \circ \cdots \circ T_1 \circ E\ (Z) \in \mathbb{R}^{N \times V},$$

where the inputs to the network $Z \in \mathbb{R}^{N \times V}$. Thus choosing a suboptimal attention function $\text{attn}(.)$ can affect every layer of the final $\text{Transformer}(.)$. Let $\ell(\text{Transformer}(Z), Y)$ be the loss function used to learn the parameters of the Transformer, where $Y$ is the true label information. Autoregressive language modeling (Radford et al., 2018; Brown et al., 2020) involves using a causal mask $M$, which is a lower triangular matrix, and is added before the softmax operation as follows:

$$\text{attn}(X) = \text{softmax}(M + QK^\top)V,$$
$$M_{ij} = 0 \text{ if } i \geq j \text{ else } -\infty$$

where $\odot$ denotes element-wise multiplication. During training, the gradients $\frac{\partial \ell}{\partial W_K}$, $\frac{\partial \ell}{\partial W_Q}$, $\frac{\partial \ell}{\partial W_V}$ are computed via backpropagation in a layer-by-layer fashion from layer $L$ to layer 1 and are used to update the parameters. In the next section, we analyze the gradient backpropagation through $\text{attn}(.)$ and propose LASER attention.

### 3.2. Gradient Analysis of Attention

For simplicity, we first let the sequence length $N$ be 2 with attention probabilities $A = \text{softmax}(QK^\top)$ and attention logits as $\tilde{A} = QK^\top$. Expanding the matrices $A$ and $\tilde{A}$, we get:

$$A = \begin{pmatrix} a_{11} & a_{12} \\ a_{21} & a_{22} \end{pmatrix} = \text{softmax} \begin{pmatrix} \tilde{a}_{11} & \tilde{a}_{12} \\ \tilde{a}_{21} & \tilde{a}_{22} \end{pmatrix}$$
$$= \begin{pmatrix} \frac{\exp(\tilde{a}_{11})}{\exp(\tilde{a}_{11}) + \exp(\tilde{a}_{12})} & \frac{\exp(\tilde{a}_{12})}{\exp(\tilde{a}_{11}) + \exp(\tilde{a}_{12})} \\ \frac{\exp(\tilde{a}_{21})}{\exp(\tilde{a}_{21}) + \exp(\tilde{a}_{22})} & \frac{\exp(\tilde{a}_{22})}{\exp(\tilde{a}_{21}) + \exp(\tilde{a}_{22})} \end{pmatrix}$$

Dividing the numerators and denominators by $\exp(\tilde{a}_{ij})$ gives:

$$A = \begin{pmatrix} a_{11} & a_{12} \\ a_{21} & a_{22} \end{pmatrix} = \begin{pmatrix} \frac{1}{1 + \exp(\tilde{a}_{12} - \tilde{a}_{11})} & \frac{1}{\exp(\tilde{a}_{11} - \tilde{a}_{12}) + 1} \\ \frac{1}{1 + \exp(\tilde{a}_{22} - \tilde{a}_{21})} & \frac{1}{\exp(\tilde{a}_{21} - \tilde{a}_{22}) + 1} \end{pmatrix}$$
$$= \begin{pmatrix} \sigma(\tilde{a}_{11} - \tilde{a}_{12}) & 1 - \sigma(\tilde{a}_{11} - \tilde{a}_{12}) \\ \sigma(\tilde{a}_{21} - \tilde{a}_{22}) & 1 - \sigma(\tilde{a}_{21} - \tilde{a}_{22}) \end{pmatrix}, \quad (3)$$

where $\sigma$ denotes the sigmoid operation $\sigma(x) = 1/(1 + \exp(-x))$. As in (2), we now multiply the attention probabilities $A$ with the value matrix $V$. For simplicity, let the representation dimension $d = 1$, then the attention output will be as follows:

$$\text{Attention output:} \ \ \text{attn}(X) = \begin{pmatrix} o_1 \\ o_2 \end{pmatrix} = AV$$
$$= \begin{pmatrix} \sigma(\tilde{a}_{11} - \tilde{a}_{12})v_1 + (1 - \sigma(\tilde{a}_{11} - \tilde{a}_{12}))v_2 \\ \sigma(\tilde{a}_{21} - \tilde{a}_{22})v_1 + (1 - \sigma(\tilde{a}_{21} - \tilde{a}_{22}))v_2 \end{pmatrix}, \quad (4)$$

where $V = \begin{pmatrix} v_1 \\ v_2 \end{pmatrix}$. We now find the gradient with respect to $\tilde{A}$ via the chain rule:

$$\underbrace{\frac{\partial \ell}{\partial \tilde{A}}}_{\text{backpropagated gradient}} = \frac{\partial \ell}{\partial \text{attn}(X)} \cdot \underbrace{\frac{\partial \text{attn}(X)}{\partial \tilde{A}}}_{\text{Jacobian}}.$$

Thus, small Jacobian magnitude can lead to small backpropagated gradient. We now analyze an element of $\text{attn}$ Jacobian:

$$\frac{\partial o_1}{\partial \tilde{a}_{11}} = v_1 \sigma(\tilde{a}_{11} - \tilde{a}_{12})(1 - \sigma(\tilde{a}_{11} - \tilde{a}_{12}))$$
$$- v_2 \sigma(\tilde{a}_{11} - \tilde{a}_{12})(1 - \sigma(\tilde{a}_{11} - \tilde{a}_{12}))$$
$$= (v_1 - v_2) \underbrace{\sigma(\tilde{a}_{11} - \tilde{a}_{12})(1 - \sigma(\tilde{a}_{11} - \tilde{a}_{12}))}_{\text{possible saturation}}. \quad (5)$$

The sigmoid function value, $\sigma(\tilde{a}_{11} - \tilde{a}_{12})$ saturates to 1 when $\tilde{a}_{11} - \tilde{a}_{12}$ becomes sufficiently large. Conversely, when $\tilde{a}_{11} - \tilde{a}_{12}$ is large and negative, the function value saturates to 0. In both cases, saturation leads to vanishing gradients, where the gradient becomes very small. This phenomenon is a well-documented limitation of the sigmoid function (LeCun et al., 2002).

We extend this observation to sequence length of size $N$ as follows:

**Lemma 3.1** (Gradient saturation in softmax). *Let $a \in \mathbb{R}^N$ be a row in attention weights/probabilities $A$ and similarly let $\tilde{a}$ be a row in attention logits $\tilde{A}$, then:*

$$\textit{forward pass:} \quad a = \text{softmax}(\tilde{a}),$$
$$\textit{backward pass:} \quad \frac{\partial \ell}{\partial \tilde{a}} = (\text{diag}(a) - aa^\top) \frac{\partial \ell}{\partial a},$$
$$\textit{softmax Jacobian:} \quad \frac{\partial a_j}{\partial \tilde{a}_i} = a_j(\mathbb{1}\{i = j\} - a_i),$$

*where* $\mathrm{diag}(a)$ *denotes the diagonal matrix with diagonal elements* $a$.

We give a proof of this lemma in Section A.2

> **Key Observation.** During the pretraining of a 2.2 billion parameter autoregressive language model, we observe in Figure 1 that about 80% of attention probabilities are less than $10^{-3}$ and about 20% are less than $10^{-7}$. From Lemma 3.1, it can be seen that small attention probabilities $a_j$, $j \in \{1, \dots, N\}$ can lead to small Jacobian values, giving diminished backpropagated gradients.

To address this issue, we now introduce LASER Attention which applies attention in exponential value space, $\exp(V)$, as follows:

$$\exp(\mathrm{laser}(X)) = \mathrm{softmax}(QK^\top)\exp(V)$$
$$\implies \mathrm{laser}(X) = \log(\mathrm{softmax}(QK^\top)\exp(V)) \quad (6)$$
$$\to \quad \text{LASER Attention,}$$

where $\log(.)$ and $\exp(.)$ are applied elementwise. Expanding (6) for $N = 2$ and $d = 1$ as done for standard attention (4) gives LASER output:

$$\alpha_1 = \sigma(\tilde{a}_{11} - \tilde{a}_{12}), \quad \alpha_2 = \sigma(\tilde{a}_{21} - \tilde{a}_{22})$$
$$\begin{pmatrix} o_1 \\ o_2 \end{pmatrix} = \begin{pmatrix} \log(\alpha_1 \exp(v_1) + (1 - \alpha_1)\exp(v_2)) \\ \log(\alpha_2 \exp(v_1) + (1 - \alpha_2)\exp(v_2)) \end{pmatrix}, \quad (7)$$

**Low gradient saturation.** Computing an element in the Jacobian $\partial\,\mathrm{laser}(X)/\partial\tilde{A}$ as done for standard attention in (5) gives the equation for LASER Jacobian:

$$\frac{\partial o_1}{\partial \tilde{a}_{11}} = \frac{(\exp(v_1) - \exp(v_2))\alpha_1(1 - \alpha_1)}{\alpha_1 \exp(v_1) + (1 - \alpha_1)\exp(v_2)}$$
$$= \frac{(\exp(v_1) - \exp(v_2))\alpha_1(1 - \alpha_1)}{\alpha_1(\exp(v_1) - \exp(v_2)) + \exp(v_2)}.$$

We now consider a limiting case to simplify the above equation: if $\exp(v_1) \gg \exp(v_2)$,

$$\frac{\partial o_1}{\partial \tilde{a}_{11}} = \frac{\alpha_1(1 - \alpha_1)}{\alpha_1 + \exp(v_2)/(\exp(v_1) - \exp(v_2))}$$
$$\approx 1 - \alpha_1 = \underbrace{(1 - \sigma(\tilde{a}_{11} - \tilde{a}_{12}))}_{\text{low saturation}},$$

where the approximation is due to $\exp(v_2)/(\exp(v_1) - \exp(v_2)) \approx 0$.

**Relation between LASER Attention and max function.** From definitions (3) and (7), LASER output can be written

in a log-sum-exp form (Blanchard et al., 2019) as follows:

$$o_1 = \log(a_{11}\exp(v_1) + a_{12}\exp(v_2))$$
$$= \log(\exp(v_1 + \log(a_{11})) + \exp(v_2 + \log(a_{12}))) \quad (8)$$

Log-exp-sum function can be thought of as a differentiable approximation of $\max$ function:

**Lemma 3.2** ((Boyd & Vandenberghe, 2004)). *The function* $f(x_1, \dots, x_n) = \log(e^{x_1} + \dots + e^{x_n})$ *is convex on* $\mathbb{R}^n$. *This function can be interpreted as a differentiable approximation of the* $\max$ *function, since*

$$\max\{x_1, \dots, x_n\} \le f(x_1, \dots, x_n) \le \max\{x_1, \dots, x_n\} + \log n$$

*for all* $x \in \mathbb{R}^n$. *(The second inequality is tight when all components of* $x$ *are equal.)*

Given that $\max(x_1, \dots, x_n)$ function is not differentiable at points where two or more elements take the same value, log-sum-exp can serve as a differentiable approximation. Using Lemma 3.2, we can relate LASER (8) to $\max(\cdot)$ operation as follows:

$$\max(v_1 + \log(a_{11}), v_2 + \log(a_{12}))$$
$$\le o_1 \le \max(v_1 + \log(a_{11}), v_2 + \log(a_{12})) + \log(2)$$

### 3.3. LASER Implementation via Log-Weighted-Sum-Exp Trick

In this section we explore implementing LASER and provide pseudocode. Given the log-sum-exp structure from (8):

$$\alpha_1 = \sigma(\tilde{a}_{11} - \tilde{a}_{12}),$$
$$o_1 = \log(\alpha_1 \exp(v_1) + (1 - \alpha_1)\exp(v_2)),$$

one can notice that $\exp(.)$ operations can lead to overflow. This problem has been recognized in (Blanchard et al., 2019) and the "log-sum-exp trick" is used to avoid overflows. However, the log-sum-exp trick cannot be applied directly as it would be difficult to implement without changing the underlying attention function. We propose a "log-weighted-sum trick", where we subtract the maximum value $m = \max(v_1, v_2)$ from $v_1$ and $v_2$ and rewrite the above equation as follows:

$$o_1 = \log((\alpha_1 \exp(v_1 - m)$$
$$+ (1 - \alpha_1)\exp(v_2 - m)) \cdot \exp(m))$$
$$= \log(\alpha_1 \exp(v_1 - m) + (1 - \alpha_1)\exp(v_2 - m)) + m.$$

Now conducting $\exp(.)$ operation on $v_1 - m$ and $v_2 - m$ will not lead to overflows. We can extend this to matrix-version

(6) by conducting column-wise maximum of value matrix $V \in \mathbb{R}^{N \times d}$ as follows:

$$m_j = \max_{i \in \{1,\ldots,N\}} V_{ij}, \, j \in \{1,\ldots,d\}$$

Define $\hat{V} \in \mathbb{R}^{N \times d}$ such that $\hat{V}_{ij} = (V_{ij} - m_j)$.

The above operations helps us conduct $\exp(.)$ operation without overflows. Then the final LASER attention operation would be as follows:

$$O = \log(\mathrm{softmax}(QK^\top)\exp(\hat{V})\mathrm{diag}(\exp(m)))$$
$$O_{ij} = (\log(\mathrm{softmax}(QK^\top)\exp(\hat{V})))_{ij} + m_j, \quad (9)$$

where $O \in \mathbb{R}^{N \times d}$. Here, $m = (m_1, \ldots, m_d)$ and $\mathrm{diag}(m)$ is a diagonal matrix with elements of $m$ as diagonals. Log-weighted-sum-exp trick allows us to implement LASER attention via merely modifying the inputs and outputs of standard attention, without changing the underlying attention function. Additionally, we show in Section 4.1 that this trick helps avoid overflows in large scale settings. The following JAX (Bradbury et al., 2018) code demonstrates that LASER attention can be implemented by utilizing standard attention functions.

```
JAX implementation of LASER
attention

# given key (B,N,H,S), value
(B,N,H,S), query (B,N,H,S)
# B - batch size, N - sequence
length
# H - number of attention heads, S
- size of the head
m = jnp.max(value, axis=1, keep
dims=True)
m = jax.lax.stop_gradient(m) # stop
the gradients along m
exp_value = jnp.exp(value - m) # shift
ing the values
f = standard_attention # attention
implementation - FlashAttention,
etc.
attention_out = f(key, query,
exp_value)
out = jnp.log(attention_out) + m #
adding back the max values
```

**Algorithm 1** LASER Attention with Log-Weighted-Sum-Exp Trick

1: **Input:** Values $V \in \mathbb{R}^{N \times d}$, Queries $Q \in \mathbb{R}^{N \times d}$, Keys $K \in \mathbb{R}^{N \times d}$
2: **Output:** LASER Attention output $O \in \mathbb{R}^{N \times d}$
3: Compute the column-wise maximum for the value matrix $V$:

$$m_j = \max_{i \in \{1,\ldots,N\}} V_{ij}, \quad j \in \{1,\ldots,d\}$$

4: Subtract $m_j$ from the $j$th column of $V$:

    // Shift values to avoid overflow in the following

    $\hat{V} \in \mathbb{R}^{N \times d}$ such that $\hat{V}_{ij} = (V_{ij} - m_j)$

5: Apply attention with Queries $Q$, Keys $K$ and Values $V$ with $m_j$, $j \in \{1,\ldots,d\}$ added back to the output, following (9)

    $O \in \mathbb{R}^{N \times d}$ is such that:

    $(O)_{ij} = (\log(\mathrm{softmax}(QK^\top)\exp(\hat{V})))_{ij} + m_j$

6: **Return** $O$

## 4. Experimental Results

### 4.1. Autoregressive Language Modeling on C4

In this section, we compare the performance of LASER Attention with standard attention mechanisms in the context of an autoregressive language modeling task.

**Dataset and Setup.** We use the C4 dataset (Raffel et al., 2020) for our experiments. The training is conducted using a batch size of 1024 sequences, each sequence has 1024 tokens. The models are trained for 160,000 iterations, resulting in the utilization of approximately 167.8 billion tokens. Throughout the training process, we monitor both the training and test losses, and we observe a significant improvement in the test set performance when using LASER Attention compared to the standard attention mechanism (as illustrated in Figure 2). We use the AdamW optimizer (Loshchilov & Hutter, 2017) paired with cosine learning rate schedule (Loshchilov & Hutter, 2016) with linear learning rate warmup followed by decay to zero at the end of the training.

**Model Architecture.** The base model architecture consists of 301 million parameters of a decoder-only Transformer, which is distributed across 32 layers as defined in (1). Each layer uses 8 attention heads, with each head having a size of 128. The MLP block in this architecture, as defined in (1), has a hidden dimension of 2048.

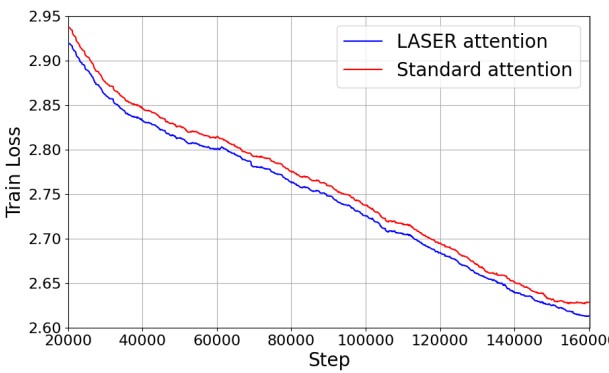

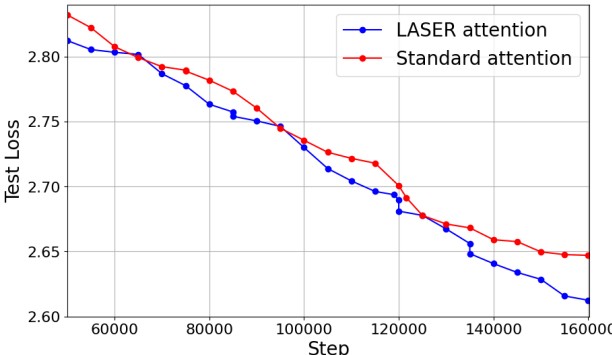

*Figure 2.* Comparison of pretraining performance between LASER and the standard attention mechanism on a 301M parameter autoregressive language model. The model consists of 32 layers, a 2048 hidden dimension, and a 1024 model dimension, trained on 168 billion tokens from the C4 dataset. LASER attention consistently achieves lower training (top) and test (bottom) loss values.

In addition to this configuration, we also experiment with a variant where the model retains 32 layers but increases the MLP block hidden dimension to 4096. In this variant, we increase the hidden dimension of the MLP block to shift more parameters into the MLP block. This configuration continues to show improvements in both the training and test loss metrics, demonstrating that the effectiveness of LASER Attention is maintained even when attention parameters are reduced. The results of these experiments can be seen in Table 1, where we also include ablation results showing improvements even with a 16-layer setting.

| Number of Layers | Hidden Dimension | LASER | Standard Attention |
|---|---|---|---|
| 16 | 4096 | **2.673** | 2.681 |
| 32 | 2048 | **2.595** | 2.641 |
| 32 | 4096 | **2.555** | 2.575 |

*Table 1.* Comparison of test loss between LASER and Standard attention mechanisms across different distribution of parameters between MLP block $f(.)$ and attention $\mathrm{attn}(.)$ in (1), where we notice upto 1.74% relative improvement in loss.

**Ablation with optimizers.** For the 301 million parameter model, we noticed in Figure 3 that LASER had higher gra-

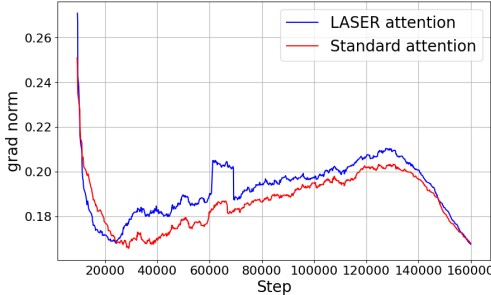

*Figure 3.* In this figure, we measure grad_norm vs steps for an autoregressive language model with a 301 million parameters model corresponding to Figure 2.

dient norm throughout training. An initial hypothesis was that higher gradient norms might lead to more parameter change, consequently reducing the loss more effectively. To investigate this, we utilized the LAMB optimizer (You et al., 2019), which normalizes and renormalizes updates using the weight norm to ensure that the scale of updates matches the scale of the weights, thus voiding the effect of gradient/update norms on optimization. Interestingly, even with LAMB's normalization mechanism, we observed a consistent improvement in training (Standard Attention: 2.749 vs LASER: 2.736) and test loss (Standard Attention: 2.758 vs LASER: 2.741), suggesting that the performance gains were not solely driven by larger gradient magnitudes but are intrinsic to the model's architecture and the LASER Attention mechanism.

**Scaling to larger models.** To demonstrate the scalability of our approach, we conducted experiments on a 1.1 billion and 2.2 billion parameter model. We note that without using the log-weighted-sum-exp trick introduced in Section 3.3, the 2.2 billion parameter model training fails due to numerical overflow. In Table 2, we show that LASER attention outperforms standard-attention in a 2.2 billion parameter model with model dimension 2048 and hidden dimension 8192 with 32 layers and 8 attention heads (each of size 512). We also train a 1.1 billion model, which has a scaled down hidden dimension (4096) and attention head size (256). We show the training curves of both the models in Figure 5, Appendix A.3. In Figure 4, we used a power law fit $f(n) = an^b$ to fit the final test loss values of LLM training runs as a function of number of parameters. Additionally, we conducted training analysis, by qualitatively measuring stability of our training runs, sensitivity to bfloat16 training in Appendix A.5.

**Performance Analysis.** We note that the 2.2B model with standard attention takes 27.22 hours on TPU v5 (Cloud, 2023) to reach a least test loss value of 2.327, however, LASER takes 24.88 hours (relative improvement of 9.4%).

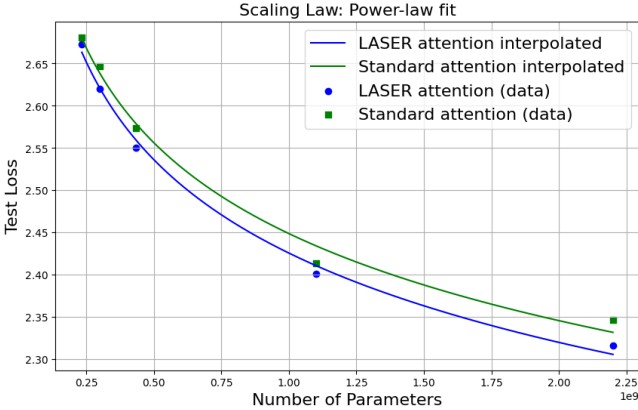

*Figure 4.* Scaling law: Power-law fit for test loss against number of parameters. This plot uses 234M, 300M, 435M, 1.1B and 2.2B parameter models' final test losses after training on $\sim 167$B tokens. To reach a loss of 2.347, it takes 15.65% fewer parameters with LASER attention.

**Baselines** Standard dot-product attention formulation in (Vaswani et al., 2017) uses a temperature of $\sqrt{d}$. Lower temperatures would be able to express sharper distributions. Attention function with temperature hyperparameter can be formalized as follows:

$$\text{Standard+temp:} \quad \text{softmax}\left(\frac{QK^\top}{\tau\sqrt{d}}\right)V,$$

$$\text{LASER+temp:} \quad \log\left(\text{softmax}\left(\frac{QK^\top}{\tau\sqrt{d}}\right)\exp(V)\right),$$

where the temperature hyperparameter $\tau$ is tuned manually. We now introduce per-dim temp operation, which scales each dimension with its own trainable temperature value:

$$\text{Standard+per-dim temp:} \quad \text{softmax}\left(\frac{QDK^\top}{\sqrt{d}}\right)V,$$

$$D = \text{diag}(\text{softplus}(p)) = \text{diag}(\log(1 + \exp(p))) \in \mathbb{R}^{d\times d},$$

where $p$ is a $d$-dimensional trainable parameter and $\text{diag}(\text{softplus}(\cdot))$ returns a diagonal matrix with positive diagonal entries to be scaled with queries. This technique is implemented in large language modeling frameworks such as PAX (Google, 2025) and AlgoPerf (Dahl et al., 2023), a framework used to compare optimizers.

In large vision transformers (Dosovitskiy et al., 2021), the norms of queries/keys can vary substantially across tokens with sharp softmax distributions in standard attention model training, leading to training instabilities. In (Dehghani et al., 2023), the authors propose QK-Normalization which conducts LayerNorm on queries and keys before computing attention weights:

Standard+QK-Normalization:

$$\text{softmax}(\text{LayerNorm}(Q)\,\text{LayerNorm}(K)^\top/\sqrt{d})V$$

**Evaluation on downstream tasks.** In Table 2, we mention the performance of our 2.2 billion parameter model on several downstream tasks. Where we evaluate on ARC (Clark et al., 2018), BoolQ (Clark et al., 2019), CB (Wang et al., 2019), COPA (Wang et al., 2019), HellaSwag (Zellers et al., 2019), MultiRC (Khashabi et al., 2018), Open-BookQA (Mihaylov et al., 2018), PIQA (Bisk et al., 2020), RACE (Lai et al., 2017), ReCoRD (Zhang et al., 2018), RTE (Wang et al., 2019), StoryCloze (Mostafazadeh et al., 2016), WiC (Pilehvar & Camacho-Collados, 2019), Winograd (Levesque et al., 2012), Winogrande (Kocijan et al., 2020), and WSC (Wang et al., 2019). We found that LASER outperforms with upto 3.38% difference and 1% difference on average in accuracy.

**(a)** Part 1

| Dataset | +per-dim temp | LASER | +per-dim temp | Standard | +temp |
|---|---|---|---|---|---|
| WSC | **81.40** | 77.89 | **79.65** | 76.14 | 78.25 |
| Winogrande | **62.27** | 61.72 | **62.35** | 60.69 | 62.19 |
| Winograd | **81.68** | 78.75 | 80.22 | **80.95** | 79.85 |
| WiC | **52.04** | 47.02 | **51.10** | 47.34 | 47.18 |
| StoryCloze | **77.87** | 77.77 | **76.48** | 76.22 | 76.17 |
| RTE | 53.43 | **54.51** | 53.07 | **54.15** | 51.70 |
| ReCoRD | **85.29** | 85.05 | **85.10** | 84.82 | 84.89 |
| RaceM | 50.42 | **50.91** | **49.65** | 49.09 | 48.89 |

**(b)** Part 2

| Dataset | +per-dim temp | LASER | +per-dim temp | Standard | +temp |
|---|---|---|---|---|---|
| RaceH | 37.88 | **37.94** | **37.65** | 37.62 | 37.04 |
| PIQA | 77.09 | **77.20** | **76.88** | 76.55 | 76.45 |
| OpenBookQA | **48.80** | 46.80 | 47.60 | 47.80 | **48.48** |
| MultiRC | 57.03 | **57.78** | 53.94 | **55.36** | 53.38 |
| HellaSwag | 66.58 | **66.70** | 65.42 | 65.53 | **65.84** |
| COPA | 82.00 | **87.00** | 80.00 | 82.00 | **85.00** |
| CB | 41.07 | **44.64** | 42.86 | 42.86 | 42.86 |
| BoolQ | 63.52 | **64.43** | 60.70 | **62.42** | 61.69 |

*Table 2.* Accuracies of one-shot evaluation of a 2.2 billion parameter autoregressive language model trained via LASER and standard attention. We found that LASER outperforms or performs the same as standard attention by up to 3.4%. On average, LASER gives an accuracy of 63.39% vs standard attention's 62.34%. We also include a temperature-tuned version of standard attention, which gives an average accuracy of 62.49%. Additionally, we report per-dimension temperature scaling for both LASER and standard attention, achieving overall means of 63.52% and 62.56% respectively.

**Finetuning on SuperGLUE.** To further evaluate LASER, we fine-tuned the 2.2B parameter model on the SuperGLUE dataset (Wang et al., 2019) for 10 epochs. These experiments provide a more comprehensive view of how LASER improves over standard attention beyond 1-shot downstream evaluations. The results, presented in Table 3, show a 1.65% average improvement in decoding accuracy for LASER over standard attention.

**Training and evaluation.** All experiments are conducted using the PAX framework (Google, 2023) built on JAX (Bradbury et al., 2018), and executed on TPUv5 chips (Cloud, 2023). We use 64 chips for 300 million param-

*Table 3.* SuperGLUE fine-tuning results for the 2.2B parameter model (decoding accuracy %). LASER shows a 1.65% average improvement over Standard Attention.

| Task | LASER | Standard |
|------|-------|----------|
| COPA | 57.00 | **58.00** |
| WiC | **56.64** | 53.92 |
| WSC | **40.38** | 36.54 |
| RTE | **22.02** | 20.94 |
| **Average** | **44.01** | 42.35 |

| Method | Valid Error | Test Error | Train Error |
|--------|-------------|------------|-------------|
| Standard | 0.2532 | 0.3749 | 0.1678 |
| Standard + temp | 0.2513 | 0.3742 | 0.1668 |
| Standard + QK-normalization | **0.2455** | **0.3644** | **0.1557** |
| Standard + per-dim temp | 0.2607 | 0.3821 | 0.1775 |
| LASER | 0.2417 | 0.3593 | 0.1527 |
| LASER + temp | 0.2427 | 0.3633 | 0.1538 |
| LASER + QK-normalization | **0.2372** | **0.3588** | **0.1427** |
| LASER + per-dim temp | 0.2475 | 0.3698 | 0.1573 |

*Table 4.* Error rates for ViT-S/16 on Imagenet. We bold the lowest errors within Standard and LASER attention variants. LASER maintains upto 1.15% improvement over standard attention variants.

Table 4 also highlights the consistent improvement of LASER over standard attention across different configurations. Temperature tuning does not introduce major changes in performance for either attention mechanism, though lower temperature values tend to degrade results due to sharper attention probability distributions. Adding QK-normalization substantially enhances performance, indicating its robustness across attention mechanisms. On the otherhand, adding per-dim temperature negatively impacts performance, likely due to hyperparameter tuning being optimized without its presence.

**Conformer on Librispeech Speech-to-Text.** We also evaluate the performance of LASER attention on the Librispeech Speech-to-Text dataset (Panayotov et al., 2015) using the Conformer model (Gulati et al., 2020). Similar to the ViT experiments, we use the AlgoPerf benchmark and perform a hyperparameter sweep across 50 configurations to optimize standard attention. We pick the optimal hyperparameters, run them for 5 different random seeds (for initialization) and report the validation curves corresponding to median in Table 5 where we demonstrate a reduction in word error rate (WER) (0.0828 → 0.0808) when using LASER attention.

eter model, 128 chips for 1.1 billion and 256 chips for 2.2 billion parameter model. Each training run takes upto 24 hours. We conducted hyperparameter search on 16-layer model mentioned in Table 1 with 15 hyperparameters using search space mentioned in Table 9 and use the optimal hyperparameter for larger models.

## 4.2. Masked Language Modeling via BERT

In the experiments so far, the focus was mainly on decoder-only models, to diversify our evaluation we now shift to encoder-only model- BERT (Devlin et al., 2018) trained via masked language modeling (as opposed to next token prediction in Section 4.1). We train a 2.2 billion parameter BERT on MLPerf training data which uses wikipedia articles (MLCommons). We used model dimension of 2048, hidden dimension - 8192, number of attention heads 16, each of size 256. We get better error rate of masked language model predictions - LASER - 0.2125 vs Standard Attention - 0.2145 (0.93% relative improvement). One can note that LASER shows more improvement in autoregressive language modeling compared to BERT.

## 4.3. Vision Transformer (ViT) and Conformer - Speech-to-Text

**Vision Transformer (ViT) on Imagenet-1k.** In this section, we experiment with the Vision Transformer (ViT) (Dosovitskiy et al., 2021) variant - S/16 on the Imagenet-1k classification task (Deng et al., 2009) which is a part of AlgoPerf benchmarks (Dahl et al., 2023) for optimizer comparisons. These benchmarks are identically implemented in init2winit framework (Gilmer et al., 2023), build on JAX, which we use for our experiments. A hyperparameter sweep was conducted over 50 configurations on NAdamW (Dozat, 2016), focusing on the search space defined in Table 8. We selected the best-performing hyperparameter configuration based on validation performance for standard attention, ran it for 5 different random seeds (for initialization) and report the corresponding training runs in Table 4, where we show that LASER attention provides a 1.15% absolute improvement in error rate (25.32% → 24.17%), i.e., a ∼4% relative improvement over standard attention.

| Method | Valid WER | Test WER | Train WER |
|--------|-----------|----------|-----------|
| Standard | 0.083192 | 0.050876 | 0.032594 |
| Standard + temp | 0.08362 | 0.050398 | 0.032275 |
| Standard + QK-Normalization | **0.082865** | **0.049747** | **0.029574** |
| LASER | 0.082892 | 0.049786 | 0.030084 |
| LASER + temp | 0.082137 | 0.049384 | 0.032822 |
| LASER + QK-Normalization | **0.080837** | **0.049231** | **0.031283** |

*Table 5.* Word Error Rate (WER) results for Conformer model with different attention mechanisms and modifications. We bold the lowest WER values within Standard and LASER attention variants. LASER shows upto 0.2% improvements on word error rates.

In Table 5, we found adding QK-Normalization enhances performance for both LASER and standard attention. Temperature tuning does not significantly impact standard attention but provides slight benefits to LASER . Since per-dim temperature is built in Conformer model architecture (Dahl

et al., 2023) we don't conduct an ablation study on its effect.

## Ablations in Large Language Modeling

In this section, we apply LASER and observe how it performs under the following changes: a) increase the model parameters to test the scalability of LASER b) using a Diff Transformer (Ye et al., 2024).

### Scalability of LASER

To test the scalability of LASER, we trained a model with 7.7B parameters for 44B tokens. Compared to the 2.2B parameter model (which had model dimension 2048, hidden dimension 8192, 32 layers, and 8 attention heads each of size 512), the 7.7B parameter model was scaled along all dimensions: model dimension 3440, hidden dimension 11584, number of heads 16, and dimension per head 720.

The downstream task evaluation for the 7.7B parameter models is presented in Table 6.

*Table 6.* Downstream task accuracies for 7.7B parameter models trained on 44B tokens. LASER shows an average improvement of 1.44% over standard attention. More pronounced differences are observed in tasks like BoolQ (+6%), CB (+1.78%), OpenbookQA (+1.8%), and RTE (+3.61%).

| Dataset | LASER | Standard |
|---|---|---|
| ArcE | 52.48 | **52.69** |
| BoolQ | **62.45** | 56.48 |
| CB | **44.64** | 42.86 |
| HellaSwag | **57.16** | 56.02 |
| MultiRC | **56.00** | 55.59 |
| OpenbookQA | **45.40** | 43.60 |
| RaceM | **44.64** | 44.15 |
| RTE | **53.43** | 49.82 |
| StoryCloze | **71.78** | 71.51 |
| WiC | **47.34** | **47.34** |
| Winogrande | **58.41** | 57.77 |
| Average | **53.97** | 52.53 |

### 4.4. Ablation with Diff Transformer

The Differential Transformer (DiffTransformer) (Ye et al., 2024) introduces a differential attention mechanism. This mechanism calculates attention scores as the difference between two separate softmax attention maps to cancel noise and promote sparse attention patterns (Ye et al., 2024). The Diff Transformer attention mechanism is formulated as:

$$\text{DiffAttn}(X) = \text{softmax}\left(\frac{Q_1 K_1^T}{\sqrt{d_k}}\right) V$$
$$- \lambda \text{softmax}\left(\frac{Q_2 K_2^T}{\sqrt{d_k}}\right) V$$

where $Q_1, Q_2, K_1, K_2 \in \mathbb{R}^{N \times d_k}$ are projected query and key vectors, $V \in \mathbb{R}^{N \times d_v}$ is the projected value vector (in the

original DiffTransformer paper, $d_v = 2d_k$ for the combined $V$), $\lambda$ is a learnable scalar, and $d_k$ is the dimension of the key/query vectors per head.

We propose a modification, LASER+DiffTransformer, by equipping the softmax dot-product attention terms with LASER modifications. The LASER attention mechanism involves conducting attention on exponentially transformed inputs, $\exp(V)$, and takes the logarithm of the result. Applying this to DiffTransformer, we get:

$$\text{DiffAttn}_{\text{LASER}}(X) = \log\left(\text{softmax}\left(\frac{Q_1 K_1^T}{\sqrt{d_k}}\right)\exp(V)\right)$$
$$- \lambda \log\left(\text{softmax}\left(\frac{Q_2 K_2^T}{\sqrt{d_k}}\right)\exp(V)\right)$$

This formulation aims to address potential backpropagation challenges due to the softmax operations in both attention maps of the DiffTransformer. We trained a 2.2B parameter DiffTransformer with Model dim: 2048, Hidden dim: 8192, Number of attention heads: 8, Head size: 512, and Number of layers: 32. The models were trained on 24 billion tokens. The results are shown in Table 7.

*Table 7.* Downstream task performance for 2.2B parameter Diff-Transformer and DiffTransformer+LASER models, trained on 24 billion tokens. On average, Diff+LASER shows an improvement of approximately 1% over DiffTransformer.

| Dataset | Diff+LASER | DiffTransformer |
|---|---|---|
| ArcE | **49.28** | 49.20 |
| CB | **42.85** | 41.07 |
| HellaSwag | **51.93** | 51.58 |
| MultiRC | **55.13** | 52.82 |
| OpenbookQA | **44.20** | 43.00 |
| RaceM | **42.40** | 40.73 |
| RTE | **52.34** | 50.90 |
| StoryCloze | 71.03 | **71.29** |
| WiC | **50.00** | 49.53 |
| Winogrande | **56.04** | 55.80 |
| Average | **51.52** | 50.59 |

## 5. Conclusion

We identified a bottleneck in the gradient backpropagation of attention mechanism where the gradients are scaled by small Jacobian values while passing through the softmax operation. We alleviate this issue by transforming the inputs and outputs of attention mechanism, and show that this leads to larger Jacobians in the limiting case. We demonstrate the improvements in training performance over four types of Transformers spanning different modalities (text, speech and vision): (a) decoder-only (via Large Language model) upto 2.2 billion parameters, (b) vision Transformers on Imagenet, (c) Conformer on Librispeech speech-to-text and (d) encoder-only (BERT) with 2.2 billion parameters,, where we show significant and consistent improvements in performance.

## Acknowledgements

We thank Devvrit Khatri, Nilesh Gupta and Aditya Kusupati for valuable suggestions.

## Impact Statement

The paper presents a technique which improves pretraining of transformers, a widely used architecture in machine learning. There are many positive potential societal consequences of our work, none which we feel must be specifically highlighted here.

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

# A. Appendix

## A.1. Hyperparameter search space

In Table 8 we outline the hyperparameter search space for all the benchmarks in Section 4.3.

| Parameter | Min | Max | Scaling/Feasible Points |
|---|---|---|---|
| learning_rate | $10^{-4}$ | $10^{-2}$ | log |
| $1 - \beta_1$ | $10^{-2}$ | 0.15 | log |
| $\beta_2$ | - | - | 0.9, 0.99, 0.999 |
| warmup_factor | - | - | 0.05 |
| weight_decay | $5 \times 10^{-3}$ | 1.0 | log |
| label_smoothing | - | - | 0.1, 0.2 |
| dropout_rate | - | - | 0.1 |

*Table 8.* Hyperparameter search space used in Section 4.3.

| Parameter | Value |
|---|---|
| learning_rate | [1e-1, 1e-2, 1e-3, 1e-4, 1e-5] |
| weight_decay | [1e-2, 1e-1, 1.0] |
| beta_1 | 0.9 |
| beta_2 | 0.99 |
| epsilon | 1e-24 |
| dropout_rate | 0.0 |

*Table 9.* Hyperparameter search space for language modeling experiments, Section 4.1

## A.2. Proofs

*Proof of Lemma 3.1.* The softmax activation function is applied row-wise on the preactivations $\tilde{A}$; we can expand this computation row-wise as follows:

$$A = \text{softmax}(\tilde{A})$$

$$\implies \begin{pmatrix} a_1^\top \\ \vdots \\ a_s^\top \end{pmatrix} = \begin{pmatrix} \text{softmax}(\tilde{a}_1^\top) \\ \vdots \\ \text{softmax}(\tilde{a}_s^\top) \end{pmatrix}$$

$$\implies a_i = \text{softmax}(\tilde{a}_i), \ \ i \in \{1, \ldots, N\}$$

$$= \left\{ \frac{\exp(\tilde{a}_{i1})}{\sum_k \exp(\tilde{a}_{ik})}, \ldots, \frac{\exp(\tilde{a}_{is})}{\sum_k \exp(\tilde{a}_{ik})} \right\}$$

$$\implies a_{ij} = \frac{\exp(\tilde{a}_{ij})}{\sum_k \exp(\tilde{a}_{ik})}$$

Taking gradient with respect to $\tilde{a}_i$ in the last expression gives:

$$\frac{\partial a_{ij}}{\partial \tilde{a}_{il}} = a_{ij}(1 - a_{ij}) \ \text{ if } l = j$$

$$= -a_{ij}a_{il} \ \text{ else}$$

Putting everything together, the Jacobian of the transformation $a_i = \text{softmax}(\tilde{a}_i)$ can be written as follows:

$$\frac{\partial a_{ij}}{\partial \tilde{a}_{il}} = (\text{diag}(a_i) - a_i a_i^\top)$$

$$\frac{\partial \ell}{\partial \tilde{a}_i} = (\text{diag}(a_i) - a_i a_i^\top)\frac{\partial \ell}{\partial a_i} \tag{10}$$

## A.3. Training loss curves

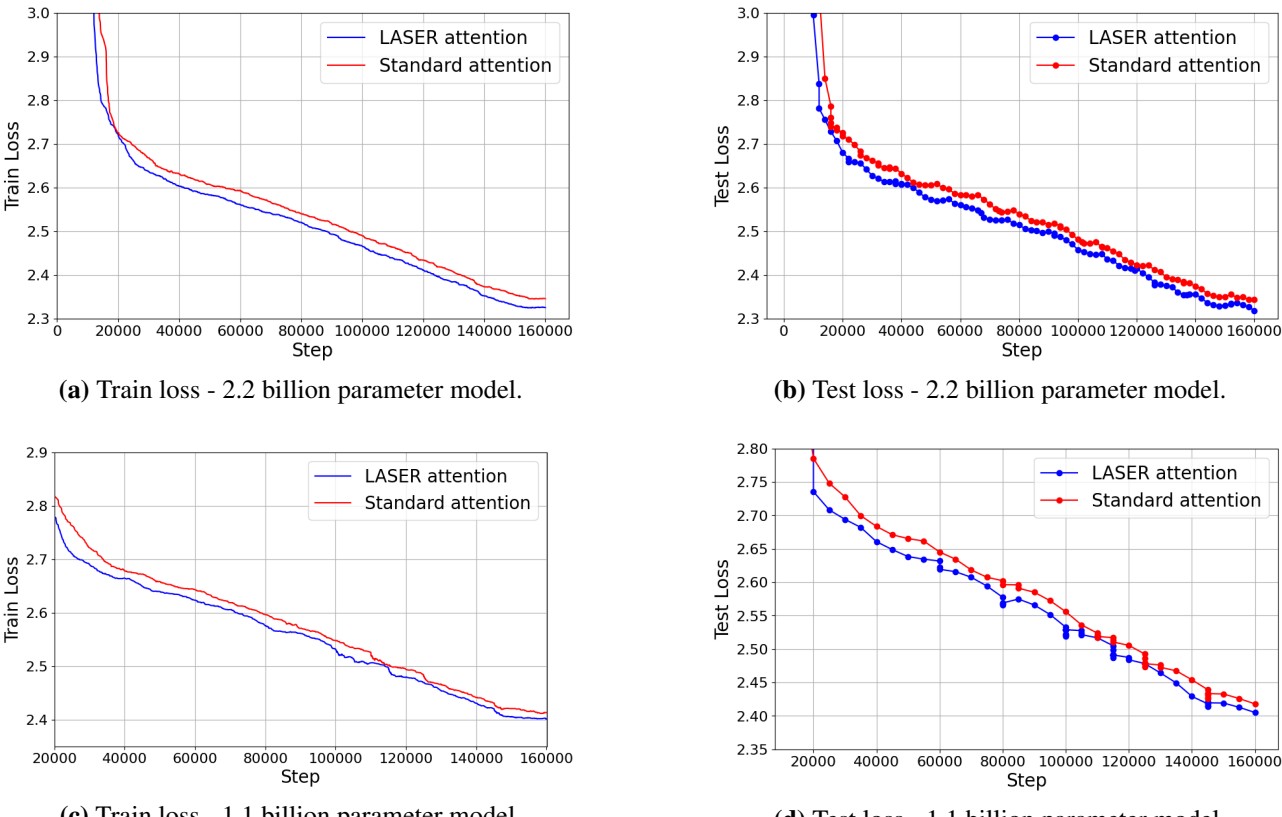

**(a)** Train loss - 2.2 billion parameter model.

**(b)** Test loss - 2.2 billion parameter model.

**(c)** Train loss - 1.1 billion parameter model.

**(d)** Test loss - 1.1 billion parameter model.

*Figure 5.* Performance comparison for 2.2 billion and 1.1 billion parameter models. The 2.2 billion model has 32 layers, 8 attention heads with head size 512, MLP hidden dimension 8192, and model dimension 2048. The 1.1 billion model has 32 layers, 8 attention heads (head size 256), MLP hidden dimension 4096, and model dimension 1024. LASER demonstrates better train and test loss compared to Standard Attention even in large-scale settings.

## A.4. Fluctuations in downstream evaluations

**(a)** Part 1

| Dataset | LASER | Standard |
|---|---|---|
| WSC | **80.98±0.68** | 79.30±0.44 |
| Winogrande | **62.19±0.21** | 62.15±0.19 |
| Winograd | **82.27±0.37** | 80.29±0.27 |
| WiC | **51.44±0.66** | 51.29±0.55 |
| StoryCloze | **77.95±0.06** | 76.46±0.07 |
| RTE | 53.14±0.14 | **53.14±0.14** |
| ReCoRD | **85.24±0.04** | 85.06±0.05 |
| RaceM | **50.56±0.18** | 49.67±0.13 |

**(b)** Part 2

| Dataset | LASER | Standard |
|---|---|---|
| RaceH | **37.99±0.13** | 37.56±0.11 |
| PIQA | **77.20±0.12** | 76.74±0.12 |
| OpenBookQA | **49.08±0.20** | 47.56±0.15 |
| MultiRC | **57.16±0.18** | 53.99±0.16 |
| HellaSwag | **66.61±0.07** | 65.46±0.04 |
| COPA | **82.00±0.00** | 80.00±0.00 |
| CB | 40.00±0.87 | **43.93±0.87** |
| BoolQ | **63.39±0.18** | 60.45±0.38 |

*Table 10.* Here we add standard deviations of each number, computed on downstream evaluations of 5 checkpoints and note an average for LASER to be 63.58±0.26 and an average for standard attention to be 62.69±0.23, noting a difference of 0.89.

## A.5. Training Analysis

**Training instability.** There can be spikes in training loss curves initially during large language model training. We notice that despite these spikes training stabilizes and converges smoothly. However, training instability/spikes can be attributed to poor model architecture and optimizer choices. We now ablate the choice of attention mechanism and understand its affect on training stability. Figure 6 compares training stability of different models.

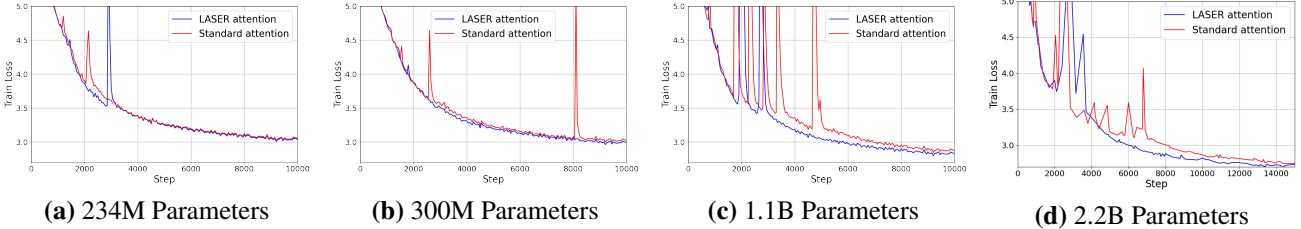

**(a)** 234M Parameters     **(b)** 300M Parameters     **(c)** 1.1B Parameters     **(d)** 2.2B Parameters

*Figure 6.* Train loss vs. steps for LASER and standard attention across different parameter scales. Training stability for each attention mechanism can be observed through the number of training spikes. Generally, models with LASER attention exhibit fewer training spikes compared to models with standard attention, indicating greater stability in training for LASER attention across all parameter scales. The figures focus on the initial part of the training as the remainder did not exhibit training instability.

Additionally, we ran pretraining of a 2.2B parameter language model presented in Section 4.1 with BF16 precision weights and activations and compare LASER and Standard attention in Table 11.

| Precision | LASER Loss | Standard Attention Loss |
|-----------|------------|-------------------------|
| **FP32**  | 2.326      | 2.344                   |
| **BF16**  | 2.333      | 2.350                   |

*Table 11.* Comparison of LASER and Standard Attention Losses for FP32 and BF16 precision on a 2.2B model.

We cached 4800 query, key, value matrices of size (1024, 8, 256) (headsize-256, heads-8, sequence length-1024) during training, and computed the following numerical reconstruction relative errors of attention output in bfloat16: vanilla LASER (0.0018, 0.0002), LASER+log-weighted-sum-exp-trick (0.0017,0.0001), Standard Attention (0.0016, 0.0001). This experiment is conducted in a machine with 4 TPUv5 chips. We conducted the same experiment on 16 A100s and found the following errors: vanilla LASER (0.002, 0.0003), LASER+log-weighted-sum-exp-trick (0.0019,0.0002), Standard Attention (0.0018, 0.0002). While on an average, we found log-weighted-sum-exp-trick to help on both TPUv5 and A100s, we note that this trick prevents overflows, which is crucial for stable training. Similar trick famously known as log-sum-exp is used to prevent overflows due to exp(.) function in the softmax function and is adopted by both Pytorch and Jax in their softmax implementations.

**Understanding training instability.** The enhanced training stability of the LASER attention mechanism can be formally attributed to its direct solution for the gradient saturation problem inherent in standard attention models. The Jacobian of the softmax function, which scales the gradients for the query and key projection matrices ($W_Q$ and $W_K$), diminishes towards zero when attention probabilities become sharp. This vanishing gradient signal create inconsistent updates to parameters, leading to suboptimal learning and potential loss spikes. LASER circumvents this by reformulating the attention output as $\log(\text{softmax}(QK^T)\exp(V))$, a structure which admits with lower saturation properties.

This improved parameter learning grants LASER its stability, a behavior analogous to residual connections in deep networks. The LASER output for a given token $i$, $o_i$, can be expressed as a log-sum-exp function, which is a tight, differentiable approximation of the $\max$ function. Here we bound $o_1$:

$$\max(v_1 + \log(a_{11}), v_2 + \log(a_{12}))$$
$$\leq o_1 \leq \max(v_1 + \log(a_{11}), v_2 + \log(a_{12})) + \log(2).$$

In cases of strong self-attention, where the attention probability $a_{ii}$ is high, the $\max$ function property ensures that the output $o_i$ closely approximates its corresponding input value $v_i$. By creating this effective "pass-through" for the input

signal during moments of high certainty, LASER functions as an inherent residual connection. We conjecture that this property is the formal mechanism that underpins the empirically observed reduction in training loss spikes compared to standard attention.

