# OpenReview forum: "LASER: Attention with Exponential Transformation"
_ICML.cc/2025/Conference — ICML 2025 poster_

### Official Review · Reviewer_o5d3 · 2025-03-05

**Overall Recommendation:** 4

**Summary:**

This paper addresses the problem of vanishing gradients in standard dot-product attention in transformers. The authors show mathematically how this problem arises in the Jacobian of the attention function during backpropagation. Next, a new technique called LASER (Logarithm of Summed Exponentials of Representations) is introduced. As the name suggests, this technique, at it's core, is like the log-sum-exp function which is a new way of formulating/approximating the weighted average in attn(.) using the log of summed exponentials. The benefit of this technique is the avoidance of the vanishing gradients as shown mathematically in this paper and consequently the better training of transformer based models.

**Claims And Evidence:**

Yes, claims are clear with convincing evidence.

**Essential References Not Discussed:**

The paper does a good job with citing the essential references.

**Experimental Designs Or Analyses:**

Yes. The experimental designs are sound and valid.

**Methods And Evaluation Criteria:**

Yes.

**Other Comments Or Suggestions:**

1. In section 3.2, the definition of $\tilde{A} = KQ^T$ is different from definition in 3.1
2. Is $v_1 - v_2 >> 0$ an obvious choice? Is it always true?
3. Equation (8) should be $log(exp(v_1 + log(a_{11})\textbf{)}+exp(v_2+log(a_{12}))\textbf{)}$ (the text is missing two closing brackets)
4. In algorithm 1, step 4: what is $exp(.)$ ?
5. In section 4.2, authors say that LASER shows more improvement in autoregressive LM compared to BERT. Is there some explanation/hypothesis to this in light of how LASER is formulated?

**Other Strengths And Weaknesses:**

Strengths:

This paper address a key issue in transformer models with a simple yet novel and innovative idea. The paper is very well written and the flow of ideas are easy to follow. The paper is also mathematically grounded with experiments spanning across multiple applications.

Weakness:

The performance improvements shown are minor.

**Questions For Authors:**

My main concern about the paper is: how significant are the improvements shown? In all the different tasks, the improvements seem minor. Did the authors perform statistical significance testing?

Furthermore, how does LASER scale to even larger transformer models? I understand that it might not be possible to conduct that experiment but the authors should comment on how LASER can be useful for larger models given how minor the improvements are.

Related to the above, does LASER introduce any computational overhead compared to vanilla attention? If yes, how is it worth using LASER given the performance and computational overhead tradeoff?

**Relation To Broader Scientific Literature:**

This paper is a key contribution to the broad literature of attention mechanisms in transformers.

**Theoretical Claims:**

I did not check the proofs.

---

> ### Author Rebuttal · Authors · 2025-04-01
>
> We appreciate the reviewer’s comments and suggestions.
>
>
> ### Addressing Other Comments or Suggestions
>
> **Comment 2:**
> Unfortunately, this is a typo — it is supposed to be `exp(v_1) - exp(v_2) >> 0`. While `v_1 - v_2` might not be significantly different, the difference between `exp(v_1)` and `exp(v_2)` can be much larger due to the exponential function, as dictated by the value parameter \( W_V \).
>
> **Comments 1 & 3:**
> Thank you for pointing these out. We will correct them in our revision.
>
> **Comment 4:**
> Thanks for pointing out this typo — there should be no `exp(.)` in that expression. We appreciate the feedback and will fix this in the revision.
>
> **Comment 5:**
> We conjecture that, since half the attention scores are zeros in decoder-only models, this could lead to less dependence on W_K and W_Q parameters, and less gradient flow through W_K and W_Q compared to BERT models.
>
> ---
>
> ### Questions for Authors
>
> > **Reviewer:** My main concern about the paper is: how significant are the improvements shown? In all the different tasks, the improvements seem minor. Did the authors perform statistical significance testing?
>
> In Table 7 (Appendix A.4), we note that the **standard deviation is low** compared to the difference between LASER and Standard attention, indicating that the observed improvements are statistically significant.
>
> > **Reviewer:** Furthermore, how does LASER scale to even larger transformer models? I understand that it might not be possible to conduct that experiment but the authors should comment on how LASER can be useful for larger models given how minor the improvements are.
>
> Please refer to our response to **Reviewer tFgw**, where we trained a **7.7B model** and demonstrated **average improvements of 1.44%**, with more pronounced gains of up to **6%** on individual downstream tasks.
>
> Additionally, please see our response to **Reviewer 9sxs**, where we fine-tuned the **2.2B model** on the **SuperGLUE dataset** [1] for 10 epochs and found a **1.65% improvement in decoding accuracy** (as opposed to eval/ranking accuracy in Table 2).
>
> > **Reviewer:** Related to the above, does LASER introduce any computational overhead compared to vanilla attention? If yes, how is it worth using LASER given the performance and computational overhead tradeoff?
>
> As noted in **Lines 326–329** under *Performance Analysis*:
> - The **2.2B model with standard attention** takes **27.22 hours** on TPU v5  to reach a minimum test loss of 2.327.
> - In contrast, **LASER takes only 24.88 hours**, a **relative walltime improvement of 9.4%**.
>
> ---
>
> [1] Wang, Alex, et al. *"SuperGLUE: A stickier benchmark for general-purpose language understanding systems."* Advances in Neural Information Processing Systems 32 (2019).

---

> > ### Comment · Reviewer_o5d3 · 2025-04-03
> >
> > I am happy with the comments and keep my score to a 4 (accept).

---

> > > ### Author Response · Authors · 2025-04-05
> > >
> > > Thank you for the response.

---

### Official Review · Reviewer_9sxs · 2025-03-12

**Overall Recommendation:** 4

**Summary:**

This paper studies the problem of Gradient saturation in softmax in the attention architecture. Thus, it proposes LASER, a log-sum-exp structure to replace the original dot-product formulation in attention with Log-Weighted-Sum-Exp Trick. Extensive experiments on different benchmarks and models, with in-depth analysis, support the development of such a novel attention mechanism.

**Claims And Evidence:**

All the claims are correct with experimental / theoretical analysis as evidence.

**Essential References Not Discussed:**

N/A

**Experimental Designs Or Analyses:**

The experimental designs and analyses are thorough, especially for the part where different optimizers are ablated.

**Methods And Evaluation Criteria:**

The method is compared with various baseline choices, adopted to different models on different tasks and is universally better than baselines on average. However, one possible concern is the improvements are marginal. Can the author justify the use of LASER in terms of other perspectives, e.g. faster convergence during training, stabilizing training, etc. While the author provides some results on the training stability in the Supplementary materials, it is appreciated if the author can provide some analysis on why LASER is stabler than original attention mechanism.

**Other Comments Or Suggestions:**

N/A

**Other Strengths And Weaknesses:**

Strengths:

1. The proposed LASER method indeed theoretically solves the gradient saturation problem.
2. The loss curves showed empirically show the effectiveness of LASER.

Weaknesses:
1. The quantitative experimental results on downstream tasks demonstrate marginal improvements compared to vanilla attention mechanism.

**Questions For Authors:**

I'm open to discussion for the above-mentioned issues, and I think the justification of LASER in terms of some analyses on the better stability could make the paper more thorough.

**Relation To Broader Scientific Literature:**

The key contribution of this paper, a new attention mechanism that aims to solve the gradient saturation problem in attention, has great potential for a great impact in attention-based models, which are dominant in the current era.

**Theoretical Claims:**

All the theoretical claims appear to be correct.

---

> ### Author Rebuttal · Authors · 2025-04-01
>
> Thank you for the reviewer’s helpful observations.
>
> > I'm curious about why LASER is stabler than vanilla attention mechanism at training time, since the author mentioned that LASER introduces larger gradient norm, intuitively this may cause fluctuation in training. Can the author provide some analyses on this from perspectives like gradient distribution?
>
> To understand this, we draw analogy to a residual connection. While residual networks add residual connections to improve gradient backpropagation, residual connections increase the gradient norm compared to vanilla network, however, the stability improves as more parameters are activated. Analogously, we conjecture that, fixing the gradient backpropagation through query and key within softmax improves the learning of W_K and W_Q.
>
>
> > I'm open to discussion for the above-mentioned issues, and I think the justification of LASER in terms of some analyses on the better stability could make the paper more thorough.
>
> We will formally prove the above in our final revision, using the aforementioned intuition.
>
> > The quantitative experimental results on downstream tasks demonstrate marginal improvements compared to the vanilla attention mechanism.
>
> In our response to Reviewer tFgw, we trained a 7.7B model and observed **average improvements of 1.44%**, with more pronounced gains of up to **6%** on individual downstream tasks.
>
> Additionally, we fine-tuned the 2.2B model on the SuperGLUE dataset [1] for 10 epochs and found a **1.65% improvement in decoding accuracy** (as opposed to eval/ranking accuracy reported in Table 2) on the SuperGLUE benchmarks listed below:
>
> | Task  | LASER | Standard |
> |-------|--------|----------|
> | Copa  | 57.00  | 58.00    |
> | Wic   | 56.64  | 53.92    |
> | WSC   | 40.38  | 36.54    |
> | RTE   | 22.02  | 20.94    |
> | **Avg** | **44.01** | **42.35** |
>
> We believe these new fine-tuning results offer a more comprehensive view of how LASER improves over standard attention, beyond what is seen in 1-shot downstream evaluations.
>
>
>
> [1] Wang, Alex, et al. *"SuperGLUE: A stickier benchmark for general-purpose language understanding systems."* Advances in Neural Information Processing Systems 32 (2019).

---

> > ### Comment · Reviewer_9sxs · 2025-04-05
> >
> > Thank the author for the point-by-point response to my questions. I appreciate the commitment the author has to prove the intuition behind using LASER's stability against vanilla attention in the final revision. Upon reading the rebuttal the authors provided to other reviewers, I'm willing to increase my score to 4 (Accept).

---

> > > ### Author Response · Authors · 2025-04-05
> > >
> > > Thank you for the reconsideration of the score.

---

### Official Review · Reviewer_8gDY · 2025-03-13

**Overall Recommendation:** 1

**Summary:**

The papers studies the gradients during backpropagation of the standard softmax dot product attention within transformers. The authors' key insight is that these gradients can be extremely small, leading to poor gradient signal propagation, which in turn leads to sub-optimal learning. They then suggest a modification of attention to mitigate this issue by suggesting that attention should be carried into in an exponential value space, involving applying an exponential function to the values matrix along with a log function to the softmax probabilites.


**Update After rebuttal:**

I thank the authors for their rebuttal. I have re-read their paper several times including the appendix. I have also read the other reviews along with their rebuttals to the other reviewers and to my review. However, I am still not convinced that this work has enough novelty and significance to be published in a conference at the level of ICML.  Therefore, I will be voting to reject the paper.

A real concern I have with the paper is that they have shown a lot of empirical results yet offer no real theoretical understanding of why their method works in the way they have reported. Furthermore, the paper has so many statement that are well known to researchers that publish in this area. For example, the whole of section 3.1 and 3.2 is well known and yet they dedicate more than a page to it. Lemma 3.1 is a very simple lemma which they claim they extend to arbitrary sequence length $N$ yet the extension is extremely simple this cannot be considered a contribution. Furthermore, the theory of the paper simply consists of computing gradients and showing that certain quantities from the computation yield a saturation term (possibly of low order). This is by no means novel or significant.

**Final Comments on Rebuttal:**

1. **We would like to highlight a subtle distinction in our approach: we apply the logarithm after the multiplication..:**

Thank you for clearing that up. I am still not convinced that this is novel. Just because an idea has not been explored that does not make it novel.

2. **Please refer to our response to Reviewer tFgw, where we scaled to 7.7B parameter models and found an average improvement of 1.44%...:**

You empirical results are fine but there is a major issue in that there is no real explanation in why you are able to get these improvements.

3. **With better gradient signal propagation, which we theoretically demonstrate in our manuscript, we conjecture that gradients may be better conditioned with LASER than with Standard attention. However, we have not yet explored the effect of this on the convergence rate of the optimizer.**

I do not agree with this entirely. You actually do not prove that your method gives better signal propagation. You simply compute gradients and show how you can then apply your method to avoid the saturation that can take place. There is no proof that what you do leads to better signal propagation and thus the relationship to the empirical results is thus still lacking. You should really prove that your method leads to better conditioned gradients. That will then clearly show your method is capping off any over flow and leading to better training.

4. **We observed similar spikes in gradient norms for both LASER and Standard attention, but only a few such spikes translated into actual loss spikes. We address this in our stability analysis (Appendix A.5), where we note that Standard attention suffers from more loss spikes than LASER.**

Thank you. I noticed massive spikes in your analysis (appendix A.5) even with normal attention which you have not made clear as to why that is happening. The fix could simply be that a better learning rate scheduler need to be chosen. Yet no analysis is done to form an actual conclusion.

5. **Thank you for pointing this out. We will explicitly mention the following limitations of LASER in our revision...**

The quadratic complexity holds even for normal attention so really you have not added anything by saying that point. I was asking about explicit limitations of LASER that are not present in usual attention. The fact that there were no limitations stated and analysed in the original paper is quite concerning, especially when reading the paper I noticed several limitations as stated above.

**Claims And Evidence:**

Yes the claims are supported by clear and convincing evidence.

**Essential References Not Discussed:**

References are fine.

**Experimental Designs Or Analyses:**

The experimental designs are fine.

**Methods And Evaluation Criteria:**

The methods and evaluation criteria are fine.

**Other Comments Or Suggestions:**

I do not have any other comments.

**Other Strengths And Weaknesses:**

**clarity:** I admire the authors efforts in trying to understand gradient propagation within the attention layer of a transformer. Their approach of mapping everything to exponential space and working there is a nice, although somewhat simple, insight. The clarity with which they write the paper is simple to read.

However, I do have the following issues:

**Novelty:** I don't feel there is enough novelty in this paper to be published in ICML. The authors simply show that gradients saturate in the attention gradients during backpropagation and then show that by applying an exponential on the values and a logarithm on the softmax probabilities one can produce a gradient that does not saturate. The approach is simply to look at the saturated gradient part and realise that a certain transformation can mitigate the effect of the saturation. Furthermore, the mathematical approach is simply just computing a derivative using the chain rule. What would have been much better is if the authors actually showed that this saturation caused serious problems with convergence of the optimizer. Or for that matter showing that by getting rid of saturation by using their LASER method that a different optimizer such as SGD can be used to train a transformer as it is well known that SGD performs extremely badly when compared to Adam on many transformers. My feeling is this will lead to nothing but the authors are welcome to prove me wrong.

**Significance:** This is related to the novelty. I don't believe the paper will be significant. While their insight is nice. It is my feeling that that is all it is. A nice insight that leads to some better performance with a simple adjustment done by carrying out some simple chain rule calculations. I feel the authors should have really shown how LASER can be impactful and significant by showing that it leads to something practitioners in the field did not expect with transformers such as better training with different optimizers like SGD.

**Questions For Authors:**

1. Would the authors be able to say something about whether theoretically LASER leads to a better convergence rate for the optimizer?
2. Furthermore, I don't really see a big difference between LASER and standard attention in the training curves. For example, in figure 2 (top curve) it seems LASER converges faster but only slightly. Since your work is that LASER should have better back propagated gradients shouldn't this lead to a much better and faster train loss curve?
3. In figure 3 I notice an issue with the gradient norm. Between step 60000 and 80000 I see that LASER has a sudden jump in gradient magnitude. This is usually an effect of gradient becoming too large which is not good for training. Did this happen in other experiments? This could be a limitation of the work.
4. What are the limitations of LASER? I noticed you did not really discuss any limitations of your method.

**Relation To Broader Scientific Literature:**

The key contributions in the paper are related to transformer training and the well known issues of gradient propagation in neural networks. this is a well studied and important topic and the authors give a new viewpoint of this.

**Theoretical Claims:**

The theoretical claims both in the paper and the appendix were checked and as best as I can see are correct.

---

> ### Author Rebuttal · Authors · 2025-04-01
>
> We appreciate the reviewer’s insightful feedback.
>
> > **Reviewer:** The authors simply show that gradients saturate in the attention gradients during backpropagation and then show that by applying an exponential on the values and a logarithm on the softmax probabilities one can produce a gradient that does not saturate.
>
> We would like to highlight a subtle distinction in our approach: we apply the **logarithm *after*** the multiplication of softmax probabilities with the values. To the best of our knowledge, this idea has **not been explored** in existing literature, and we believe it offers a **novel attention formulation**.
>
> ---
>
> > **Reviewer:** Significance: This is related to the novelty. I don't believe the paper will be significant.
>
> Please refer to our response to **Reviewer tFgw**, where we scaled to **7.7B parameter models** and found an **average improvement of 1.44%** in downstream evaluations, with **more pronounced improvements up to 6%** in individual tasks. Additionally, we show that this insight **generalizes to other tasks** such as:
> - **ViT: +1.2%**
> - **BERT**
> - **Speech Transformer**
>
> Also, in our response to **Reviewer 9sxs**, we fine-tuned the **2.2B model on the SuperGLUE dataset** [1] for 10 epochs and observed a **1.65% improvement in decoding accuracy** (in contrast to the eval/ranking accuracy reported in Table 2) across the SuperGLUE benchmarks.
>
> ---
>
> > **Reviewer:** Would the authors be able to say something about whether theoretically LASER leads to a better convergence rate for the optimizer?
>
> With better gradient signal propagation — which we theoretically demonstrate in our manuscript — we **conjecture** that gradients may be **better conditioned** with LASER than with Standard attention. However, we have **not yet explored** the effect of this on the convergence rate of the optimizer. We will consider including this analysis in our final revision.
>
> ---
>
> > **Reviewer:** In Figure 3 I notice an issue with the gradient norm. Between step 60000 and 80000 I see that LASER has a sudden jump in gradient magnitude. This is usually an effect of gradient becoming too large which is not good for training. Did this happen in other experiments? This could be a limitation of the work.
>
> We observed similar spikes in **gradient norms for both LASER and Standard attention**, but only **a few such spikes translated into actual loss spikes**. We address this in our **stability analysis (Appendix A.5)**, where we note that **Standard attention suffers from more loss spikes than LASER**.
>
> ---
>
> > **Reviewer:** What are the limitations of LASER? I noticed you did not really discuss any limitations of your method.
>
> Thank you for pointing this out. We will explicitly mention the following **limitations of LASER** in our revision:
> - LASER attention has **quadratic computational complexity**, which can be significant for long sequence lengths.
> - A **comprehensive failure case study** for LASER (and similarly for Standard attention) has not been conducted. This remains **future work**.
>
> ---
>
> [1] Wang, Alex, et al. *"SuperGLUE: A stickier benchmark for general-purpose language understanding systems."* Advances in Neural Information Processing Systems 32 (2019).

---

### Official Review · Reviewer_tFgw · 2025-03-14

**Overall Recommendation:** 2

**Summary:**

This paper introduces LASER (LogArithm of Summed Exponentials of Representations), a novel attention mechanism for Transformers. The researchers found that in the standard attention mechanism, the gradients backpropagated through the softmax operation can be small, which may lead to inefficient learning of parameters. LASER addresses this by applying attention in the exponential value space. It conducts attention on exp(V) and uses a Log - Weighted - Sum - Exp trick to avoid numerical overflow. Experiments on various Transformer models, including autoregressive LLMs, Vision Transformer, and Conformer, show that LASER can improve performance. For example, in autoregressive language modeling on the C4 dataset, it can achieve up to a 1.74% relative improvement in test loss, and it also shows better results in downstream tasks with an average accuracy improvement of about 1%.

**Claims And Evidence:**

yes

**Essential References Not Discussed:**

no

**Experimental Designs Or Analyses:**

yes

**Methods And Evaluation Criteria:**

yes

**Other Comments Or Suggestions:**

no

**Other Strengths And Weaknesses:**

Strengths
1. LASER really shows its worth in experiments. It can significantly reduce the test loss in autoregressive language modeling, like up to 1.74% in some cases. And it also improves the accuracy in downstream tasks, which means it can make the model perform better in practical applications. For instance, in the Vision Transformer on Imagenet, it has a 1.15% absolute improvement in error rate, which is a great result.
2. It's really convenient that LASER can be implemented with just small modifications to the existing attention implementations. It doesn't need to change the underlying attention function, which makes it easy for researchers and engineers to apply in their work. This simplicity also means it can be quickly integrated into different models.
3. The paper provides a solid theoretical analysis. By studying the gradient backpropagation in the attention mechanism, it clearly points out the problem of small gradients in the standard attention and explains how LASER can solve this problem. This theoretical support makes the proposed method more reliable and convincing.
4. LASER has been tested on a variety of models across different modalities, such as text, speech, and vision. It shows consistent improvements in all these models, which indicates that it is a versatile method and not limited to a specific type of model or task. This broad applicability makes it more valuable in different research and application scenarios.

Weaknesses
1. Although LASER shows good results in many experiments, it may not be suitable for all situations. The paper mainly focuses on models with a certain scale of parameters, and it's not clear how well it will perform in models with extremely small or large numbers of parameters. Also, it might face challenges in some specific tasks where the data characteristics are very different from those in the experiments.
2. The Log - Weighted - Sum - Exp trick is crucial for LASER to avoid numerical overflow. But this also means that the performance of LASER is highly dependent on this trick. If there are some issues with this trick in different hardware or software environments, it may affect the performance of LASER. And the paper doesn't fully explore the potential problems that might occur when using this trick.
3. The paper mainly compares LASER with the standard attention mechanism and some simple modifications of it. There are many advanced attention mechanisms proposed recently, and the paper doesn't compare LASER with them. So, it's hard to say how LASER stands out among the most advanced techniques in the field. This lack of comparison limits the understanding of LASER's superiority and competitiveness.

**Questions For Authors:**

see weaknesses

**Relation To Broader Scientific Literature:**

no

**Theoretical Claims:**

yes

---

> ### Author Rebuttal · Authors · 2025-04-01
>
> We thank the reviewer for their comprehensive comments.
>
> >  The paper mainly focuses on models with a certain scale of parameters, and it's not clear how well it will perform in models with extremely small or large numbers of parameters.
>
> We trained a model with **7.7B parameters scaling up by 3.5x our largest model (2.2B)**. We train the model for 44B tokens and evaluate it on downstream tasks:
>
> | Dataset     | Laser  Acc | Standard  Acc |
> |-------------|----------------|--------------------|
> | arc_e       | 52.48          | 52.69              |
> | boolq       | 62.45          | 56.48              |
> | cb          | 44.64          | 42.86              |
> | hellaswag   | 57.16          | 56.02              |
> | multirc     | 56.00          | 55.59              |
> | openbookqa  | 45.40          | 43.60              |
> | racem       | 44.64          | 44.15              |
> | rte         | 53.43          | 49.82              |
> | storycloze  | 71.78          | 71.51              |
> | wic         | 47.34          | 47.34              |
> | winogrande  | 58.41          | 57.77              |
> | **average** | **53.97**      | **52.53**          |
>
> We found that compared to 2.2B billion model which gave an gain of 1% on average, for 7.7B model, we found about **1.44% average gain over standard attention**, and observe **more pronounced differences in boolq (+6%), cb (+1.78%), openbookqa (+1.8%), rte (+3.61%)**.  In 7.7B parameter model, we scaled along all dimensions, model_dimension=3440, hidden_dimension=11584, num_heads=16, dims_per_head=720.
>
> We conducted a power-law fit (Figure 4) on test loss of models ranging from 234M to 2.2B and conjecture that **smaller models might have smaller differences in loss value with LASER**. However, this observation is **for decoder language models on C4**. In contrast, our experiments with **ViT-S/16 with 22M parameters, LASER gives 1.2% improvement** in classification error.
>
> > Also, it might face challenges in some specific tasks where the data characteristics are very different from those in the experiments.
>
> While we showed gains on standard downstream tasks used to evaluate the performance large language models, identifying tasks where LASER performs poorly and similarly softmax-attention performs poorly is interesting future work we plan to do.
>
>
> > The Log - Weighted - Sum - Exp trick is crucial for LASER to avoid numerical overflow. But this also means that the performance of LASER is highly dependent on this trick.
>
>
> **We cached 4800 query, key, value matrices of size (1024, 8, 256) (headsize-256, heads-8, sequence length-1024) during training**, and computed the following numerical reconstruction relative errors of attention output in bfloat16: **vanilla LASER (0.0018, 0.0002),  LASER+log-weighted-sum-exp-trick (0.0017,0.0001), Standard Attention (0.0016, 0.0001)**. This experiment is conducted in a machine with 4 **TPUv5 chips.** We conducted the same experiment on **16 A100s** and found the following errors:  **vanilla LASER (0.002, 0.0003),  LASER+log-weighted-sum-exp-trick (0.0019,0.0002), Standard Attention (0.0018, 0.0002)**. While on an average, we found log-weighted-sum-exp-trick to help on both TPUv5 and A100s, **we note that this trick prevents overflows, which is crucial for stable training**. Similar trick famously known as **log-sum-exp [1] is used to prevent overflows due to exp(.) function in the softmax function** and is adopted by both pytorch and jax in their softmax implementations.
>
> >  There are many advanced attention mechanisms proposed recently, and the paper doesn't compare LASER with them.
>
> **Diff Transformers [1] is a recent state of the art attention and concurrent work, which outperforms Standard Attention** which uses the difference of two softmax attention matrices and their multiplication with the value matrix as the output. We note that both matrices still face backpropagation challenges due to the use of softmax. To address this, we use the LASER formulation. We trained a 2.2B DiffTransformer with Model dim: 2048, Hidden dim: 8192, Number of attention heads: 8, Head size: 512, Number of layers: 32. The model was trained on 24 billion tokens.
> | Dataset     | Diff+LASER | DiffTransformer |
> |-------------|-------------|-----------------|
> | arc_e       | 49.2845     | 49.2003         |
> | cb          | 42.8571     | 41.0714         |
> | hellaswag   | 51.9319     | 51.5834         |
> | multirc     | 55.1361     | 52.8259         |
> | openbookqa  | 44.2000     | 43.0000         |
> | racem       | 42.4095     | 40.7382         |
> | rte         | 52.3466     | 50.9025         |
> | storycloze  | 71.0315     | 71.2988         |
> | wic         | 50.0000     | 49.5298         |
> | winogrande  | 56.0379     | 55.8011         |
> | **Average** | **51.5235** | **50.5951**     |
> On average, **we observe an improvement of ~1%, similar to the improvement in 2.2B transformer model in Section 4.1**.
>
> [1] Ye, Tianzhu, et al. "Differential transformer."  (2024).

---

### Decision · Program_Chairs · 2025-05-01

**Decision:**

Accept (poster)

**Comment:**

I believe the paper is worth accepting. While LASER's innovation in addressing gradient backpropagation through attention may be modest, the empirical evidence is compelling. However, I agree that wider field adoption would require more extensive practical validation beyond what's shown in the paper. The authors have responded well to reviewer concerns with additional experiments on larger models and diverse tasks. Those are a major factor that I consider when I recommend the acceptance.